# CGM-Freq: A Python Library for Frequency Domain Analysis of Continuous Glucose Monitoring Data

Elizabeth Healey and Isaac Kohane

*Abstract*—In recent years, the number of patients using continuous glucose monitoring (CGM) has increased. In addition to helping patients manage their disease, CGM produces time series data that can be used for integration in control algorithms, predictive models, and for retrospective analyses. Through feature extraction, many digital biomarkers can be derived from CGM. In this work, we provide a tool to extract features derived from the frequency domain. We first introduce a novel open-source Python library, CGM-Freq, for the analysis of CGM data in the frequency domain. We then test the library on real data. This work provides an open-source tool to further investigate the frequency domain of CGM signals.

*Index Terms*—signal processing, continuous glucose monitoring

## I. INTRODUCTION

Continuous glucose monitors (CGM) are commonly used wearable devices among patients with diabetes that collect continuous data on interstitial glucose levels [1]. Increasingly, there has been interest in analyzing retrospective CGM data to improve clinical outcomes through building predictive models and identifying associations between CGM features and clinical outcomes [2]–[4]. Much of this work has leveraged feature extraction from raw CGM data.

While there have been many recent advances in computational tools for analyzing CGM data [5], there has been limited work focusing specifically on frequency-domain analysis of CGM data. In this work, we present a new open-source tool in Python that can be used when analyzing CGM data in the frequency domain. We discuss the components of the library and test it on open-source data.

Our primary contributions are:

1) We provide an easy-to-use tool to visualize CGM signals in the frequency domain using advanced signal processing techniques.
2) We provide a library with functions for the generation of tabular digital biomarkers from the CGM data. The feature generation from time series data includes features derived from the frequency domain and features derived from the time domain. These features can be used when building predictive models or analyzing retrospective data.
3) We test the library on publicly available CGM data. Through this analysis, we confirm the potential of features from the frequency domain to discriminate different disease states.

## II. RELATED WORK

### A. Existing packages to analyze CGM data

Over the past decade, there have been many different computational tools developed for the analysis of CGM data [6]–[13]. These tools exist in multiple programming languages including R, Python, MATLAB, and through online graphical user interfaces (GUI), and they provide plots and feature derivation from raw CGM data. One of the limitations of the currently available tools is that they do not include a comprehensive frequency domain analysis.

### B. Previous work supporting frequency domain analysis of CGM

There has been recent interest in analyzing the frequency spectrum of CGM signals. Fourier analysis of CGM has been explored in recent literature, and the results of which have suggested that there may be insights to be gained from analyzing the frequency domain of CGM signals [14]–[17]. In Fico et al., the authors analyzed the CGM signals from individuals and showed that frequency-domain features were associated with the different diagnoses of T2D, T1D, and individuals at risk for T2D. Features from the frequency domain have also been incorporated in predictive models [18], further supporting the rationale for developing more analysis tools.

Further, in the broader landscape of physiological signal processing, there has been demonstrated interest in characterizing the frequency spectrum of biological signals, and computational resources exist for this purpose [19], [20]. However, these tools are not specialized for CGM data.

## III. DESIGN AND IMPLEMENTATION

### A. Overview

CGM-Freq is a library for analyzing CGM features in the frequency domain. The toolbox takes in raw CGM input from a file with required headers for columns representing the time and the CGM values.

The library has four primary modules:

Elizabeth Healey is with the Harvard-MIT Program in Health Sciences and Technology, Cambridge, MA, USA ehealey@mit.edu. Isaac Kohane is with the Department of Biomedical Informatics, Harvard Medical School, Boston, MA, USA. This work was supported by the National Science Foundation Graduate Research Fellowship program under grant 2141064.
The Python library is currently available online at https://github.com/lizhealey/CGM-Freq.

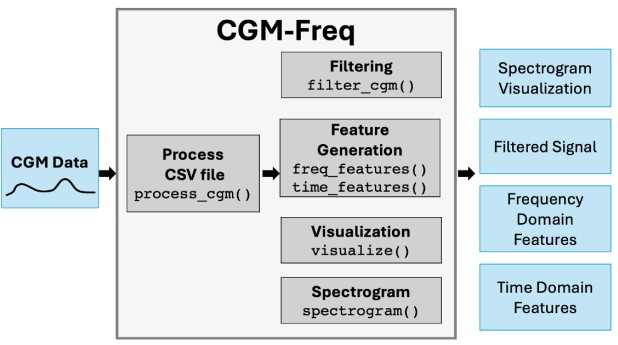

Fig. 1. Overview of the CGM-Freq library

.

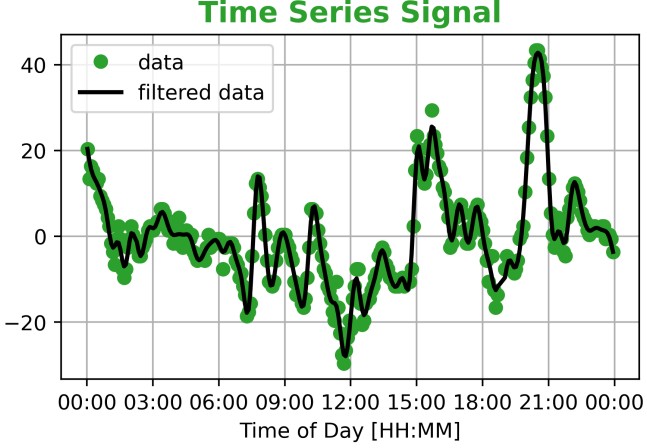

Fig. 2. Filtered CGM signal. This plot shows the output of the filter function with the filtered signal in black
.

1) **Filtering**. This module takes in the CGM signal and applies a low-pass filter based on a cutoff frequency that can be adjusted by the user. It returns and plots the filtered signal.
2) **Feature Generation**. This module has two parts. The first part computes frequency-domain features and the second part computes basic time-domain features.
3) **Visualization**. This module allows for the visualization of frequency domain of the signal and plots the time series signal.
4) **Spectrogram**. This module returns the output of the spectrogram and plots the spectrogram along with the time series signal.

Fig. 1 gives an overview of the system and the functions for each module.

### B. Preprocessing

The CGM signal first must be preprocessed using a function called *process_cgm()*. The output of this function is two arrays: an array of the CGM values and an array of the time values. A warning is also printed making note of how many CGM readings are estimated to be missing based on the sampling frequency. In the current version of our library, we do not impute missing data.

### C. Filtering

The CGM data is filtered using a low-pass filter. The input to the function includes the time and CGM values and the cutoff frequency, which can be adjusted. In this function, a filtered CGM signal is returned and plots are produced for visualization. Fig. 2 shows the visualization that is produced by this function. The raw CGM data is plotted with the filtered data.

### D. Feature Generation

Features are generated using two different functions. Table I shows the features generated and their definitions. The feature generation functions are split by frequency domain features and time domain features.

The first function generates features from the frequency domain. These features are derived based on previous work

[17] which extracts frequency-domain features from the signal using advanced signal processing techniques [21], [22]. In this function, the Discrete Fourier Transform is computed using the fast Fourier transform (FFT) algorithm. Additionally, the power spectral density (PSD) is generated using the Welch method [23]. Here, the window type and the number of samples per segment are input variables that can be modified to change the frequency resolution of the PSD.

The second function captures the time-domain features of the signal. We draw from metrics described in literature for CGM interpretation [24]. Since the purpose of this package is frequency analysis of CGM signals, we do not include a comprehensive list of time-domain features that have been previously reported. Both functions produce tabular data, enabling a quick transformation of the time series data for downstream tasks.

### E. Visualization

The *visualize()* function takes in raw CGM data and returns a plot with three components. Fig. 3 shows the output of the function. The plots contain the FFT, the PSD, and the raw CGM data. The function also takes in the sampling frequency and optionally takes in additionally arguments for the PSD. The PSD is obtained using the *welch()* from SciPy [25]. While only 24 hours of data are shown, the function is capable of plotting longer time periods of time.

### F. Spectrogram

This part of the toolbox produces both a visual output and numerical outputs of the spectrogram transformation from SciPy [25]. In Fig. 4, the visual output of the spectrogram function is shown. The function also returns the raw output of the spectrogram transformation, for analysis of the power spectrum as a function of time at different frequencies. The visual output can be used to identify temporal trends in the frequency domain of the signal.

| Frequency Domain Features | |
| --- | --- |
| *Discrete Fourier Transform* | |
| **Feature** | **Definition** |
| max_amplitude | The maximum magnitude of the FFT [17] |
| fft_dominant_frequency | The frequency where the maximum of the FFT occurs [17] |
| fft75_frequency | The frequency where 75% of the FFT signal is contained [17] |
| fft_peak2_frequency | The frequency where the second highest peak of FFT occurs [17] |
| fft_peak2_mag | The magnitude of the FFT where the second highest peak of the FFT occurs [17] |
| *Power Spectral Density* | |
| **Feature** | **Definition** |
| psd_max_amplitude | The maximum value of PSD [17] |
| psd_dominant_frequency | The frequency where the maximum value of PSD occurs [17] |
| bandwidth | The 3dB frequency bandwidth [17] |
| psd75_frequency | The frequency where 75% of the PSD is contained [17] |
| **Time Domain Features** | |
| **Feature** | **Definition** |
| Mean | Mean glucose in the segment |
| Minimum | Minimum glucose value in the segment |
| Maximum | Maximum glucose value in the segment |
| TIR | The percent of readings in the range 70mg/dL to 180mg/dL in the segment |
| TAR 1 (>180) | The percent of readings above 180 mg/dL in the segment |
| TAR 2 (>250) | The percent of readings above 250 mg/dL in the segment |
| TBR 1 (<70) | The percent of readings below 70 mg/dL in the segment |
| TBR 2 (<54) | The percent of readings below 54 mg/dL in the segment |
| Std | The standard deviation of glucose readings in the segment |
| CV | The coefficient of variation of glucose readings in the segment |
| | *The features and definitions for the frequency domain features are drawn from definitions in [17] and the features and definitions for the time domain features are drawn from [24]* |

## G. Methods

We leverage several existing Python libraries including NumPy [26], SciPy [25], and Matplotlib [27]. Specifically, we leverage the *find_peaks()* function from SciPy. The full list of dependencies for the library are listed in the Github repository. The publicly available data used in this work was made available under a Creative Commons Attribution License [28].

## IV. DEMONSTRATION ON REAL DATA

### A. Testing Dataset

We used a public dataset from Hall et al. [28] that has CGM data from individuals with T2D, with pre-diabetes, and with no diabetes. We used CGM-Freq to extract features from daily CGM data for each individual in the dataset. When analyzing the data, we included only CGM data where there were greater than 280 readings in the day.

### B. Frequency domain features on real data

To analyze the features, we plotted the mean and standard deviation of the features by diagnosis group. Fig. 5 shows a visual representation of the separation of means for each of the features by diagnosis group. Many of the time-domain features and frequency-domain features had different mean values for each diagnosis group. As expected, the mean average glucose was higher in individual with diabetes than patients without diabetes.

In Table II, the mean and standard deviation of each feature is shown for each diagnosis group. The total number of individuals and CGM days is also shown in the table. A t-test was performed to assess significance in differences in the mean between the "No diabetes" group and the "pre-diabetes" group, and between the "No diabetes" group and the "diabetes" group. Many of the features had statistically significant different means. These results align with previous findings from Fico et al, [17] which found statistically significant differences in values of many frequency domain features, such as the maximum peak of the PSD, between groups of individuals with T2D and individuals at risk of diabetes.

We further analyzed the CGM features across the individuals to understand the heterogeneity. We took the mean value of each CGM feature for each individual and plotted the absolute value of the correlation coefficients of the features. Fig. 6 shows how the CGM features are correlated with one another. Both the fft75_frequency and psd75_frequency have correlation coefficients less than .7 with all time-domain features, but statistical significance in distinguishing groups based on diagnosis. This suggests that the frequency domain features may provide unique insight into glycemic dynamics.

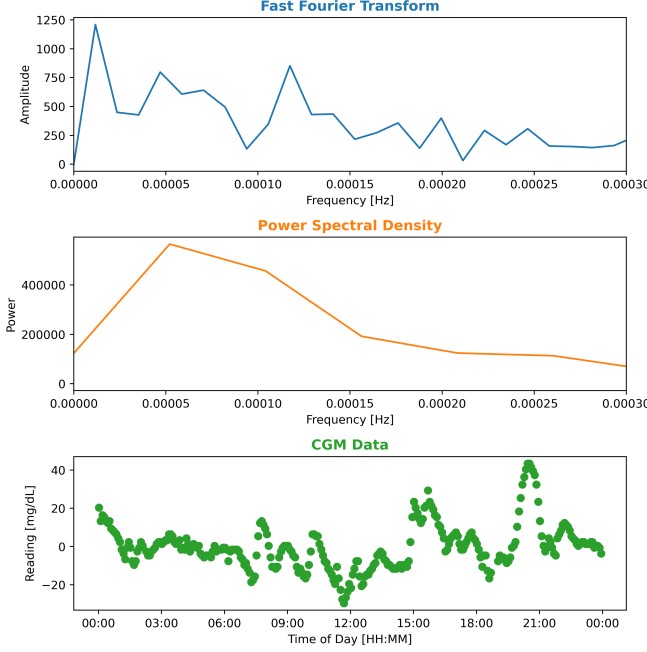

Fig. 3. Visualization output. These plots show the FFT (top), PSD (middle) and raw CGM data (bottom) for 24 hours of sample data.

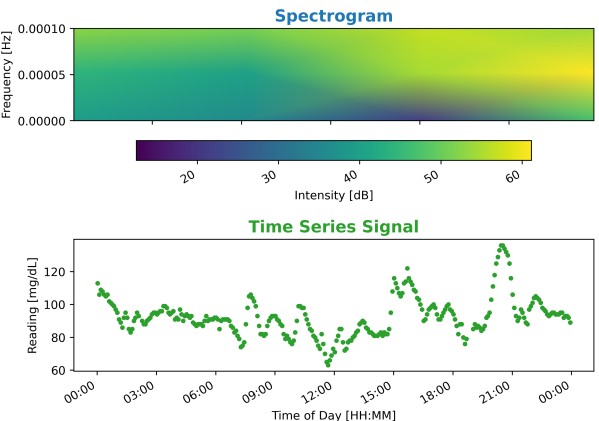

Fig. 4. Spectrogram of CGM data. These plots show a sample of 24 hours of CGM data with the spectrogram (top) and the raw time series data (bottom). The x axis is time of day for both plots.

## V. DISCUSSION

In this work, we introduce the first version of a Python library for generation of digital biomarkers from CGM signals in the frequency domain. In our library, we have four modules: visualization, filtering, feature generation, and spectrogram generation. This library serves as an open-source tool for analysis of CGM data in the frequency domain.

The library was tested on a public dataset of patients with diabetes, pre-diabetes, and no diabetes. The results of the testing were presented in the paper, confirming the discriminative power of multiple frequency-domain features, which is

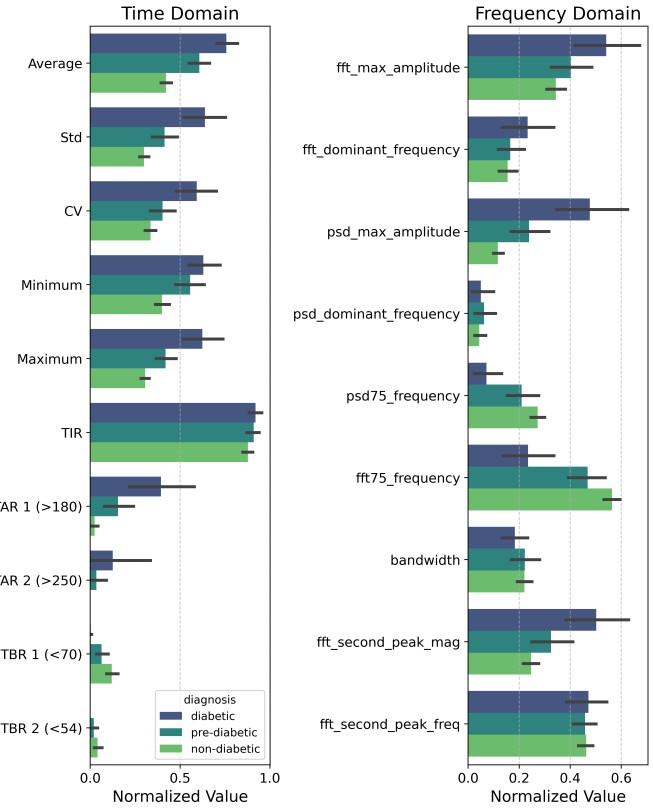

Fig. 5. CGM features by diagnosis. The values for each feature are normalized within each feature for demonstrative purposes. The mean is shown, with error bars indicating the standard deviation.

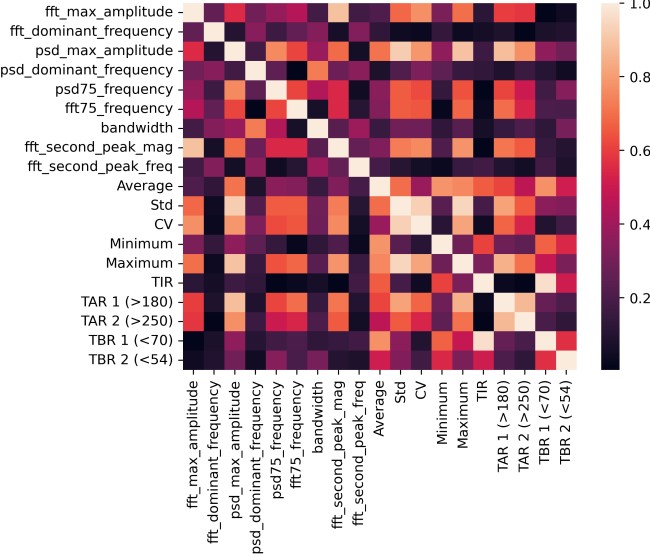

Fig. 6. Heatmap showing the absolute value of the correlation coefficients of the features across all individuals. For individuals with data from multiple days, we take the mean feature value.

consistent with previous literature [17].

The data this was tested on contained only a small number of individuals with limited heterogeneity in disease phenotype.

TABLE II
FEATURE VALUES AND SIGNIFICANCE

| | No diabetes | Pre-diabetes | Diabetes |
|---|---|---|---|
| Number of individuals in category | 25 | 10 | 2 |
| Number of days included | 132 | 46 | 12 |
| *Time Domain Features* | | | |
| Average | 96.30 (10.31) | 107.96 (13.08)* | 117.35 (7.15)* |
| CV | 0.18 (0.05) | 0.19 (0.06)* | 0.25 (0.06)* |
| Maximum | 154.98 (27.12) | 179.31 (41.01)* | 222.92 (43.73)* |
| Minimum | 61.77 (10.94) | 69.93 (14.95)* | 73.75 (8.44)* |
| Std | 17.10 (5.29) | 21.11 (8.09)* | 28.95 (7.52)* |
| TAR 1 (>180) | 0.00 (0.01) | 0.02 (0.03)* | 0.05 (0.04)* |
| TAR 2 (>250) | 0.00 (0.00) | 0.00 (0.01)* | 0.01 (0.01)* |
| TBR 1 (<70) | 0.06 (0.09) | 0.03 (0.06)* | 0.00 (0.00)* |
| TBR 2 (<54) | 0.00 (0.01) | 0.00 (0.01) | 0.00 (0.00) |
| TIR | 0.93 (0.11) | 0.94 (0.07) | 0.95 (0.04) |
| *PSD Features* | | | |
| bandwidth | 3.8e-05 (2.7e-05) | 3.9e-05 (3.2e-05) | 3.2e-05 (1.8e-05) |
| psd75_frequency | 1.4e-04 (3.8e-05) | 1.2e-04 (5.4e-05)* | 8.7e-05 (2.3e-05)* |
| psd_dominant_frequency | 4.2e-05 (2.0e-05) | 4.6e-05 (2.4e-05) | 4.3e-05 (1.6e-05) |
| psd_max_amplitude | 2.0e+06 (1.8e+06) | 4.0e+06 (4.0e+06)* | 7.8e+06 (4.2e+06)* |
| *FFT Features* | | | |
| fft75_frequency | 3.7e-04 (7.3e-05) | 3.3e-04 (1.0e-04)* | 2.3e-04 (7.2e-05)* |
| fft_dominant_frequency | 1.7e-05 (1.9e-05) | 1.8e-05 (1.6e-05) | 2.4e-05 (1.7e-05) |
| fft_max_amplitute | 3.3e+03 (1.8e+03) | 3.8e+03 (2.3e+03) | 5.0e+03 (2.0e+03)* |
| fft_second_peak_freq | 1.4e-05 (4.0e-05) | 1.3e-05 (4.0e-05) | 1.7e-05 (3.8e-05) |
| fft_second_peak_mag | 2.5e+03 (1.4e+03) | 3.2e+03 (2.3e+03)* | 4.6e+03 (1.9e+03)* |
| *\* indicates p<.05 in t-test with "no diabetes" group* | | | |

We expect that the discriminative power of the features may be different in a larger cohort. Our library provides a tool to include frequency-domain features in future analyses of CGM data. Below we highlight some potential use cases of our library:

1) Predictive models: Predictive models from time series data are often built using extracted features in tabular form. This library provides an easy-to-use feature extraction function to quickly transform raw CGM data into features that can be used to build predictive models.
2) Anomaly detection: By studying how frequency components change over time, it may be possible to identify anomalies in the data.
3) Unsupervised clustering: Digital biomarkers that char-

acterize CGM signals in the frequency domain can be included in clustering approaches to identify novel phenotypes.

CGM-Freq is a library for researchers to quickly transform CGM data into time and frequency domain features in Python. Our library serves as an additional computational resource that investigators can use when analyzing CGM data. The toolbox will continue to be updated and expanded to include more features as research in this space progresses.

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
