# OpenReview forum: "CGM-Freq: A Python Library for Frequency Domain Analysis of Continuous Glucose Monitoring Data"
_IEEE.org/EMBS/BHI/2024/Conference — IEEE BHI'24_

### Official Review · Reviewer_bTGr · 2024-07-29
**Applicable for analyzing continuous glucose data, but its validation and efficacy need further investigation.**

**Overall Rating:** 7
**Confidence:** 3

**Other Quality Metrics:**

Clarity of writing: Great
Clinical Significance: Good
Methodological Novelty: Good
Experiments Results: Fair

**Questions For The Authors:**

-	Reference 21 should be modified to an identifiable work
-	Regarding the feature extraction for predictive models, how does the CGM-Freq library ensure the reliability and relevance of the features extracted from raw CGM data for use in predictive models?
-	How effective is the CGM-Freq library in identifying digital biomarkers for understanding disease heterogeneity, and what specific advantages does it offer over existing methods, considering the frequency domains analysis?

**Strengths:**

-	The paper introduces a novel Python library for frequency domain analysis of CGM data, filling a gap in existing tools.
-	The library uses advanced techniques like FFT and PSD for detailed CGM signal analysis.
-	 The tool effectively distinguishes different glycemic statuses using real-world data.
-	 The open-source library is a valuable resource for researchers, promoting further CGM data analysis.

**Summary Of The Paper:**

The paper "CGM-Freq: A Python Library for Frequency Domain Analysis of Continuous Glucose Monitoring Data" introduces an open-source Python package designed for the frequency domain analysis of continuous glucose monitoring (CGM) data. This tool enables researchers to extract and analyze digital biomarkers from CGM data using advanced signal processing techniques, such as Fast Fourier Transform (FFT) and Power Spectral Density (PSD). The library was tested on a public dataset, demonstrating its capability to identify significant differences in frequency domain features among individuals with different glycemic statuses (no diabetes, pre-diabetes, and diabetes).

**Weaknesses:**

The current version requires users to manually validate data quality, which can be a cumbersome process.

---

### Official Review · Reviewer_tZPC · 2024-08-12
**Review of CGM-Freq**

**Overall Rating:** 7
**Confidence:** 5

**Other Quality Metrics:**

The clarity of writing is great. The clinical significance is excellent. The methodological novelty is good and the results are good as well.

**Questions For The Authors:**

In Fig. 2, how do the authors define the time axis? You mentioned the CGM device samples glucose every 5 min, but I can't catch that in this figure. Please make the time axis a bit clear or explain it in the caption. In Fig. 3 (bottom), you showed it better.

**Strengths:**

The paper is well written. The authors clearly describe the contributions of their paper. The authors properly reviewed the related works. The authors clearly described different parts of the package and presented their results.

**Summary Of The Paper:**

The paper is about introducing a python package using frequency domain analysis of glucose monitoring data.

**Weaknesses:**

- Please put spaces between a word and parantheses for defininng an abbreviation such as defining T2D.
- Please put space after the dot of end of each sentence and the next sentence if they are in one paragraph such as in Section I: "... diabetes [5].Much ..."
- In Section III-C: please state the value 0.0005 properly, not .0005.
- Please use "Fig. X" to mention Figure X in the text as it is in the figure captions.
- In Table III: the authors should revise the  dominant frequency and psd_dominant frequency to match the writing style.
- In Fig. 4, if I am right, the x axis of the top figure is Time. Can you please write the x axis for the top figure? This enhances the readability of this figure.
- Please use a separate section for References.

---

### Official Review · Reviewer_P7we · 2024-08-19
**CGM-Freq: A Python Library for Frequency Domain Analysis of Continuous Glucose Monitoring Data**

**Overall Rating:** 5
**Confidence:** 3

**Other Quality Metrics:**

Clarity of writing - good
Clinical Significance - fair
Methodological Novelty - poor
Experiments and Results - poor

**Questions For The Authors:**

In section 3, the authors describe: "This module takes in the CGM signal and applies a low-pass filter based on a desired cutoff frequency." How this cut off can influence the results? This is a very subjective point in the analysis that can comprise the next steps of the processing.

The manuscript represents a description of a toolbox, but not a scientific contribution.

The Feature List presented in Table III was based on the literature? Please, provide more refs to support.

**Strengths:**

The impact of those solutions can be huge, and very positive for a large group of patients worldwide.

**Summary Of The Paper:**

The paper presents a tool for continuous glucose monitors (CGM). The authors use several features to differentiate between patients without diabetes, those with pre-diabetes, and those with diabetes. The authors perform a statistical analysis to confirm that are differences between the three groups of patients. A more robust analysis should be performed.

**Weaknesses:**

Scientific contribution is poor. A study of the metrics assessed and the changes with characteristics from the patients would increase the impact of the study.
Several sentences are vague, such as: "Many of the features had statistical significantly different means."

---

### Decision · Program_Chairs · 2024-09-23

Accept